evolution, ecology, plant science

life-history trade-offs, climate gradient, snow removal, fitness components, fruit abortion, phenotypic and genetic correlations

**Author for correspondence:**
Jill T. Anderson
e-mail: jta24@uga.edu

# Costs of reproduction under experimental climate change across elevations in the perennial forb *Boechera stricta*

Elena Hamann[1], Susana M. Wadgymar[2] and Jill T. Anderson[1,3]

[1]Department of Genetics and the Odum School of Ecology, University of Georgia, Athens, GA 30602, USA
[2]Biology Department, Davidson College, Davidson, NC 28035, USA
[3]The Rocky Mountain Biological Laboratory, Crested Butte, CO 81224, USA

EH, 0000-0003-2888-6440; SMW, 0000-0001-6503-9799; JTA, 0000-0002-3253-8142

Investment in current reproduction can reduce future fitness by depleting resources needed for maintenance, particularly under environmental stress. These trade-offs influence life-history evolution. We tested whether climate change alters the future-fitness costs of current reproduction in a large-scale field experiment of *Boechera stricta* (Brassicaceae). Over 6 years, we simulated climate change along an elevational gradient in the Rocky Mountains through snow removal, which accelerates snowmelt and reduces soil water availability. Costs of reproduction were greatest in arid, lower elevations, where high initial reproductive effort depressed future fitness. At mid-elevations, initial reproduction augmented subsequent fitness in benign conditions, but pronounced costs emerged under snow removal. At high elevation, snow removal dampened costs of reproduction by prolonging the growing season. In most scenarios, failed reproduction in response to resource limitation depressed lifetime fecundity. Indeed, fruit abortion only benefited high-fitness individuals under benign conditions. We propose that climate change could shift life-history trade-offs in an environment-dependent fashion, possibly favouring early reproduction and short lifespans in stressful conditions.

## 1. Introduction

Investment in one life function comes at the expense of other functions, which generates life-history trade-offs under finite resources [1]. For example, allocating resources to current reproduction can reduce resources needed for future survival, growth and fecundity [2–5] (electronic supplementary material, figure S1). These costs of reproduction underlie the evolution of life-history strategies that range from semelparity, in which individuals have a single fatal reproductive event, to iteroparity, in which reproductive effort is distributed over a longer lifespan to optimize lifetime fitness [6,7].

Empirical studies have investigated costs of reproduction across diverse taxa [8–11], but often yield ambiguous results, in part owing to methodological differences that affect the magnitude of costs and the ability to detect these costs [12–14]. Within a population, individuals can vary in how they allocate limited resources (e.g. to current reproduction versus future fitness) [2–5], and how effectively they acquire limiting resources, such that some individuals have a greater offspring production across their lifespans [15,16]. The expression of life-history trade-offs depends on the abiotic conditions individuals experience [12,13], which determine resource availability [16]. When resources are abundant, individuals can compensate for allocation to reproduction through continuous or increased resource uptake, thereby dampening the costs of reproduction [12,13,17]. Additionally, high resource availability under benign or stable environments could eliminate costs of reproduction altogether, or even lead to positive associations between current reproduction and future fitness [15,18–20]. Such positive correlations could arise as an artefact of microenvironmental variation,

Proc. R. Soc. B 288: 20203134

as microsites with greater resource availability could augment both current reproduction and future fitness. However, these positive correlations could also reflect genotypic variation in resource acquisition rates [15,16]. Resource limitation can intensify the costs of reproduction, as allocation to reproduction can reduce the resources available for growth and survival [13]. For example, in plants, increased costs of reproduction can occur under limited water availability [21,22], elevated temperatures [23] and reduced soil fertility [24], and costs can vary with light conditions [25,26] and growing season lengths [27]. Critically, we know little about how rapid climate change could influence this key life-history trade-off [28].

Contemporary climate change exposes natural communities to increased temperatures, altered precipitation and more frequent and severe extreme events such as heatwaves, droughts and floods [29]. These changing conditions can reduce resource availability and impose novel selection that can alter resource allocation and life-history strategies [30,31]. For iteroparous species, climate change could shift the distribution of reproductive effort across the lifespan, cause resource allocation to diverge from optimal allocation strategies, heighten existing costs of reproduction or generate new costs under stressful conditions. For example, experimental warming increased the cost of reproduction in one of four alpine species [23], and costs also increased in dry environments in a perennial primrose [22], and long-lived orchid [28]. However, relatively few studies directly link costs of reproduction to natural or experimentally manipulated climate change factors; thus, the effects of rapid climate change on life-history trade-offs remain poorly understood [28]. Consequently, a central question as climate change continues, is whether reproduction under novel—often stressful—conditions could increase fitness costs, by depleting the reserves necessary for individuals to survive and reproduce in future years [32].

Empirical studies typically examine the costs of reproduction by quantifying how current reproductive effort affects future fitness components, such as survival, growth and fecundity [13]. Yet, studies rarely account for individuals that initiate reproduction but fail to produce viable offspring; this reproductive failure consumes resources that could have been allocated to growth or survival. Flower and fruit abortion occurs frequently in plants [33,34], and could be an adaptive strategy to cope with variation in environmental conditions [35]. Plants can compensate for uncertain biotic environments by producing excess flowers early in the growing season, and selectively aborting fruits if the environment becomes too stressful for successful seed development. If individuals continue to grow during reproduction or can recover and reallocate resources from aborted structures to other life-functions, fruit abortion could be adaptive [12,33,35]. However, few studies have empirically tested whether resources are recovered and reallocated following fruit abortion [36,37]. Instead, the considerable resources allocated to floral and fruit development could be lost when fruits fail to mature [12,35], and overproduction of fruits and subsequent fruit abortion could exaggerate costs of reproduction, and represent a maladaptive response to environmental variation.

Here, we test the hypothesis that novel, stressful climates could exacerbate costs of reproduction for populations of *Boechera stricta* (Brassicaceae), a perennial forb native to the Rocky Mountains. This alpine habitat is dominated by iteroparous perennials with slow life cycles [38–40]. We predict

that climate change could shift the costs of reproduction and drive the evolution of alternative life-history strategies in this system. To examine the costs of reproduction, we manipulated snow dynamics in five common gardens arrayed across a broad elevational gradient. Our snow removal treatment simulates climate change conditions in this region by accelerating snowmelt, decreasing soil water availability and lengthening the growing season [41]. We examined the direction and magnitude of phenotypic and genetic correlations between the first-year reproductive effort (successful and failed reproduction) and future fitness components, including survival, the probability of reproduction and fecundity. By evaluating both genetic and phenotypic correlations, our analyses accounted for microenvironmental variation in resource availability within each common garden to expose fitness trade-offs that arise at the genotypic level.

We tested whether: (i) costs of reproduction are greater in arid low elevation sites compared with mesic high elevation locales, and (ii) snow removal augments the costs of reproduction within each garden. As climate change progresses, high elevation populations will experience warmer and drier conditions similar to current conditions at lower elevations [41,42] and we predict intensifications of costs of reproduction and shifts in the distribution of the reproductive effort across the lifespan in higher elevation sites. Finally, we tested if (iii) fruit abortion is an adaptive strategy to reallocate resources or a maladaptive response to changes in environmental conditions.

## 2. Material and methods

### (a) Study system

*Boechera stricta* (Graham) Al-Shehbaz (Brassicaceae) is a short-lived perennial forb native to the Rocky Mountains, where it ranges in elevation from 700 to 3900 metres above sea level (m.a.s.l.) and latitudes from Utah to Alaska [43]. We conducted our study in the Colorado Rocky Mountains, where populations maintain high genetic variation [44,45]. This species is primarily self-pollinating [46], making it ideal to study the effects of abiotic conditions on costs of reproduction, as fecundity does not depend on biotic interactions with pollinators.

### (b) Experimental design

We established five common gardens near the Rocky Mountain Biological Laboratory (Gothic, Colorado) at 2553, 2710, 2890, 3133 and 3340 m.a.s.l., reflecting the local elevational distribution of *B. stricta* [41]. In this region, temperature declines, snow melts later and soil moisture increases with elevation [47,48]. Prior to the experiment, we collected seeds from plants separated by at least 1.5 m in natural populations that spanned a broad elevational gradient (2694–3690 m.a.s.l.), and grew them in the greenhouse for one generation to homogenize maternal effects and generate full-sibling families. In October 2013, we transplanted three-month-old plants from those full-sibling families into 60 × 120 cm blocks containing natural vegetation in each garden (*n* = 7094, from 104 maternal families and 43 source populations, of which *n* = 5824 survived the first winter and were included in analyses; *n* = 14–16 blocks, except in the highest garden, which had seven blocks). To minimize microenvironmental effects on fitness, we randomized one full sibling per family into each block and alternated snow removal and control blocks in each garden. Snowmelt timing, soil temperature and soil moisture varied minimally across blocks of the same treatment within each elevation and our multiyear experiment captured inter-annual climate variation representative of historical

climates (tables S1–S3 and fig. S1 in [41]). Transplanting in the autumn exposed plants to winter cues necessary for flowering. We measured the rosette diameter as a metric of initial size [41]. From planting to 2019, $n = 11$ individuals died owing to experimenter error and $n = 705$ died owing to indiscriminate gopher tunnelling during the winter [49] and a gopher breach of the fence at one elevation (3133 m). We retained these individuals as right-censored data in analyses of the longevity costs of reproduction (see statistical analyses), but excluded them from models of the fecundity costs of reproduction, as premature mortality precluded quantification of lifetime fitness (figure 1c).

In the Rocky Mountains, climate change decreases winter snowpack and hastens spring snowmelt [42]. Reduced snow-water equivalent and increased evapotranspiration from warming temperatures enhances drought stress during growing seasons [41,50]. From spring 2014–2019, we exposed half of the experimental individuals to ambient snow dynamics, and the other half to early snow removal. By shovelling snow to a depth of 10 cm when snowpack receded to 1 m (March–June, depending on the elevation), we advanced snowmelt by an average of 5.7–19.1 days and reduced soil water availability [41].

## (c) Data collection

We monitored survival, flowering success and fecundity from April 2014 to October 2019 by visiting all individuals 2–3 times per week. For reproductive individuals, we measured fecundity by summing the length of all mature siliques (fruits) for each individual (a reliable proxy for seed set [51]). We also recorded the number of siliques that successfully matured and the number that failed to set seed to quantify fruit loss (failed reproduction).

## (d) Statistical analyses

To test whether simulated climate change conditions exacerbate costs of reproduction, we used generalized linear models that analysed how reproductive effort in the first year (successful and failed reproduction) affected three fitness components: longevity (plant lifespan), future probability of reproduction and future fecundity. We calculated first-year fitness as the total mature silique length produced by each individual in 2014, with values of zero for individuals that produced no mature fruits. We quantified first-year failed reproduction as the total number of failed siliques of each individual in 2014. We examined our three response variables as a function of first-year fitness, failed reproduction, garden elevation (a categorical fixed factor) and snow removal treatment, and all two- and three-way interactions of first-year fitness or failed reproductive effort with garden and treatment. Negative effects of first-year fitness or failed reproduction on future fitness components indicate a cost of initial successful or failed reproduction, whereas positive effects signify a benefit of reproduction or suggest that fruit abortion is adaptive. Significant interaction terms reflect shifts in costs of reproduction across abiotic environments.

We standardized first-year fitness and failed reproduction to a mean of 0 and standard deviation of 1. All models included initial size as a covariate and random effects for maternal family and block to account for genotypic differences and microenvironmental variation. Previous analyses from this and complementary experiments documented local adaptation to elevation and geographical location of source populations [41,49]. To account for local adaptation, we included elevational difference (source elevation—garden elevation) and geographical distance between source populations and transplant gardens as covariates.

### (i) Longevity costs

A mixed-effects Cox proportional hazards model (R coxme package v. 2.2-16, [52]) evaluated longevity costs of first-year fitness and failed reproduction, using the aforementioned fixed and random effects. Hazards ratios greater than 1 or less than 1 indicate that high initial reproduction increased or decreased mortality, respectively (i.e. reduced or increased lifespan). Plants that did not die during the course of the experiment were right-censored at the last census date [53]. We used the full dataset for this analysis, as individuals that died from gopher activity or experimenter error were right-censored at the last observation point.

### (ii) Fecundity costs

We investigated fecundity costs in hurdle models that account for zero-inflation. First, we analysed how first-year fitness and failed reproduction affected the cumulative probability of reproducing in all subsequent years using a mixed model logistic regression with the aforementioned fixed and random effects (glmer function of lme4 R package, v. 1.1-23, [54]). We assigned individuals that reproduced at least once from 2015 to 2019 a score of 1 and those that failed to reproduce a score of 0. In the second part of the hurdle model, we examined future fecundity costs of initial reproduction among individuals that successfully reproduced at least once after the first growing season. We quantified future fecundity as the sum of mature silique length from 2015 to 2019, excluding individuals that did not reproduce in those years. We used mixed model regression with a gamma distribution and log link (glmer function of lme4 R package, v. 1.1-23, [54]), and determined the significance of fixed effects with Type III sums of squares from the car package v. 3.0-7 in R [55] and of random effects via likelihood ratio tests. Our power to detect costs of reproduction at the lowest elevation (2553 m) was severely restricted by the limited number of individuals that reproduced after the first growing season (figure 1c). Therefore, we excluded the lowest elevation garden from fecundity analyses.

### (iii) Lifetime costs

To examine the lifetime consequences of allocating resources to failed fruits, we analysed lifetime fecundity as a function of the proportion of fruits that failed in the first year (failed fruits/total fruits), garden elevation, treatment and all interactions, with the aforementioned covariates and random effects in a generalized linear mixed model with a gamma distribution and log link (glmer function of lme4 R package, v. 1.1-23, [54]). We calculated lifetime fecundity as the sum of mature silique length from 2014 to 2019 for individuals that flowered in at least one growing season, excluding individuals that never reproduced. We included all gardens in this analysis.

To evaluate the combined effect of first-year fitness and failed reproduction, we analysed lifetime fecundity as a function of initial silique length, initial failed silique number, garden elevation, treatment and all 2-, 3- and 4-way interactions with a covariate for initial size and random effects for block and genotype in a generalized linear mixed model with a gamma distribution and log link (glmer function of lme4 R package) [54]. We visualized significant interactions between failed and successful reproduction using the plotLMER3d.fnc function of the LMERConvenienceFunctions R package [56].

### (iv) Genetic correlations

We calculated genetic correlations between initial reproduction and subsequent fitness components using generalized linear mixed models in the R package MCMCglmm v. 2.26 [57]. These genetic correlations test genotypic variation in resource allocation and resource acquisition across environments [13]. We modelled $2 \times 10^6$ Markov chain Monte Carlo (MCMC) iterations with the first 30% of iterations ($6 \times 10^5$) discarded for burn-in and 2000 iterations sampled for estimating the mean and 95% credible intervals of genetic correlations within each garden elevation individually, for control and removal treatments [45]. These Bayesian models use MCMC methods to fit non-Gaussian data [57]. We extracted genetic correlations from two multivariate models that jointly

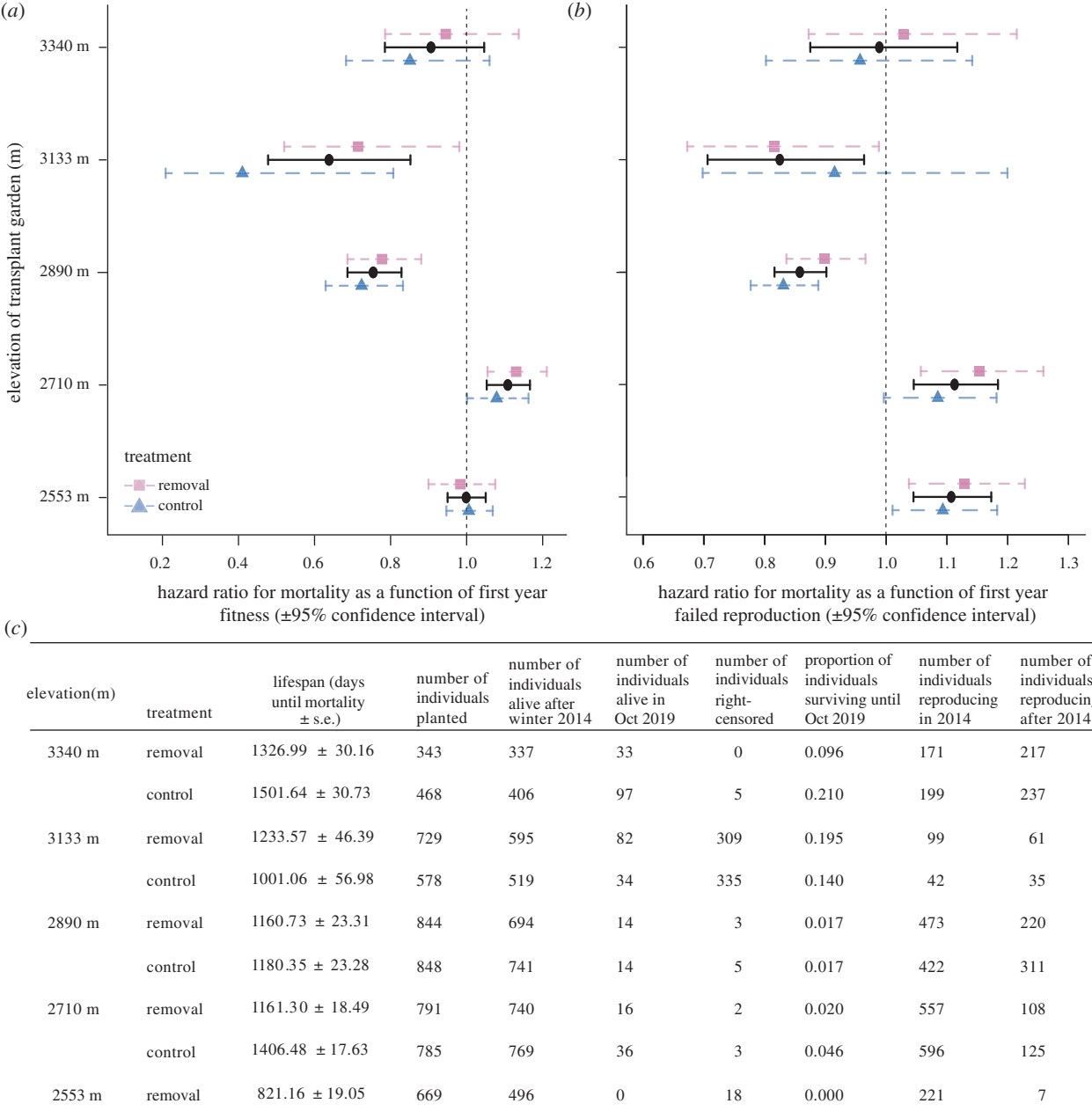

**Figure 1.** Hazard ratio of mortality and 95% confidence intervals from Cox proportional hazards models as a function of first-year fitness (*a*) and failed reproduction (*b*) across garden elevations. Hazards ratios greater than 1 indicate that high first-year fitness or failed reproduction increased mortality (i.e. reduced lifespan) and values less than 1 show that high first-year fitness or failed reproduction depressed mortality (i.e. increased lifespan). (*c*) Sample sizes and the average lifespan for individuals that survived the first winter. To calculate lifetime and the proportion of individuals surviving, we excluded transplants that died from gopher mortality and experimenter error. (Online version in colour.)

examined future fitness (probability of reproduction after 2014 or fecundity after 2014) and initial reproduction (first-year successful and failed reproduction). From these multivariate models, we estimated bivariate genetic correlations between initial and future fitness, and between initial failed reproduction and future fitness, as the genetic covariance between initial and subsequent fitness components divided by the square root of the product of the genetic variance of each fitness component. One multivariate model failed to converge (snow removal at 3340 m for post-2014 fecundity); therefore, we conducted separate bivariate analyses of post-2014 fecundity versus first-year fitness, and versus first-year failed reproduction. We specified random effects for genotype (equivalent to total genetic variance, $V_g$) and block effects using *us* covariance structure [57]. We tested whether genetic correlation estimates significantly differed between environments by evaluating overlap in the 95% credible intervals.

## 3. Results

### (a) Longevity

Excluding plants that died during the first winter and those destroyed by gopher tunnelling, experimental individuals lived an average of 3.23 years ($1182.1 \pm 189.8$ days) and 7.5% of the transplants survived until October 2019 (figure 1*c*). Thus, our 6-year long experiment captured the full lifespan of most plants. The significant interaction between garden elevation and first-year fitness indicated that the longevity cost of reproduction varied across elevations ($\chi^2_4 = 82.3$, $p < 0.0001$; electronic supplementary material, table S1). We found no influence of first-year fitness on longevity at the highest and lowest elevation gardens, but a pronounced longevity cost of reproduction emerged at the second-lowest garden

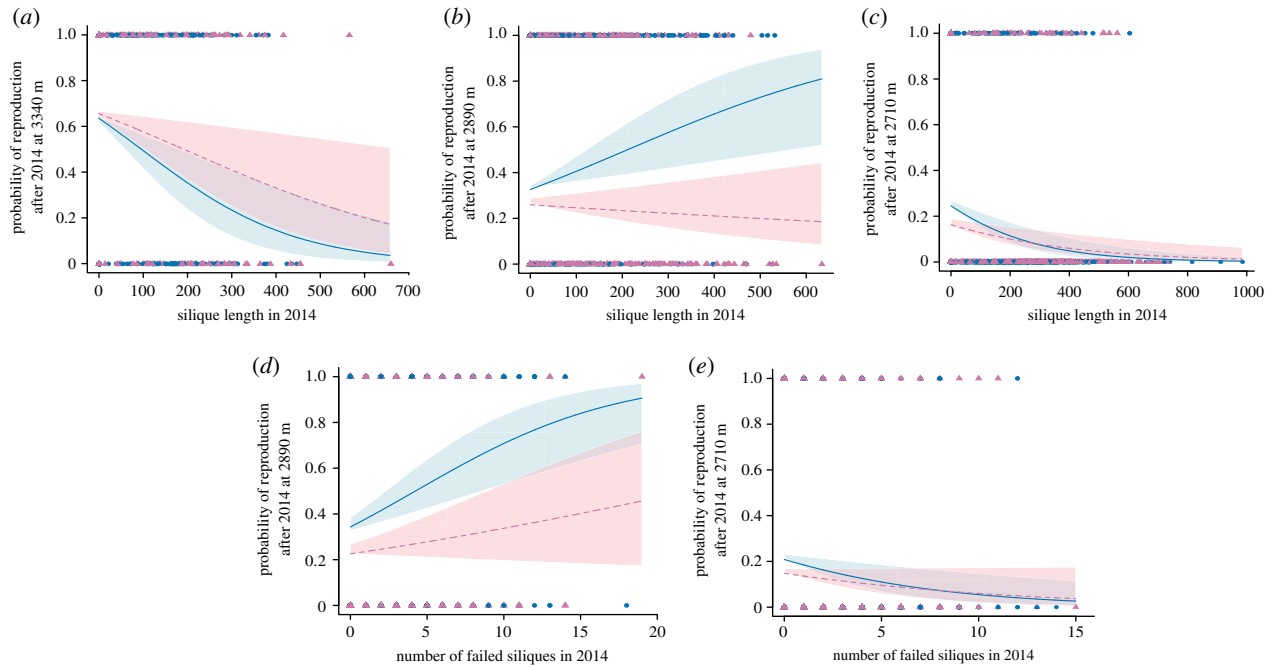

**Figure 2.** Probability of reproducing after 2014 as a function of first-year fitness (silique length in 2014: *a–c*), and of first-year failed reproduction (number of failed siliques in 2014: *d,e*) at common garden elevations with significant slopes. Panels show predicted relationships from mixed-effect logistic regression for the probability of reproducing after 2014 with 95% confidence intervals derived from $n = 2000$ bootstraps, in blue for control and pink for snow removal (printed version: solid lines and light grey for control, and dashed lines and dark grey for snow removal). Electronic supplementary material, figure S2 contains a version of figure 2 with histograms displaying the distribution of the data for each panel. (Online version in colour.)

(2710 m) (figure 1*a*; electronic supplementary material, table S2). By contrast, at mid-elevation gardens (2890 and 3133 m), greater first-year fitness lengthened lifespan (figure 1*a*; electronic supplementary material, table S2).

The significant interaction between garden elevation and first-year failed reproduction ($\chi^2_4 = 65.7$, $p < 0.0001$; electronic supplementary material, table S1) occurred because failed reproduction severely reduced longevity at the lowest elevation gardens (2553 and 2710 m), increased longevity at mid-elevations (2890 and 3133 m) and did not influence longevity at the highest elevation (3340 m, figure 1*b*; electronic supplementary material, table S2).

## (b) Probability of reproduction
The probability of reproduction after the first season varied as a function of a significant three-way interaction among first-year fitness, garden elevation and treatment ($\chi^2_3 = 17.37$, $p = 0.00059$; electronic supplementary material, table S3). We found a significant cost of first-year fitness on future odds of reproduction at high and low elevations (3340 and 2710 m, respectively) in both treatments (figure 2*a,c*; electronic supplementary material, table S4). By contrast, the probability of subsequent reproduction increased with first-year fitness in the control treatment at the mid-elevation (2890 m), but snow removal entirely eliminated this benefit of initial reproduction (figure 2*b*; electronic supplementary material, table S4).

Garden elevation and experimental treatment influenced the costs of first-year failed reproduction (number of failed siliques in 2014 × garden elevation: $\chi^2_3 = 28.47$, $p < 0.0001$, and number of failed siliques in 2014 × treatment: $\chi^2_1 = 5.29$, $p = 0.021$; electronic supplementary material, table S3). At mid-elevation (2890 m), the future probability of reproducing increased with first-year failed reproduction under control

conditions (figure 2*d*; electronic supplementary material, table S5). By contrast, at low elevation (2710 m), high initial failed reproduction lowered future reproductive odds (figure 2*e*; electronic supplementary material, table S5). First-year failed reproduction had no effect on the future probability of reproducing for all other garden elevation by treatment combinations (electronic supplementary material, table S5).

## (c) Fecundity among individuals that successfully reproduced after the first year
The fecundity costs of first-year fitness differed across garden elevation by treatment ($\chi^2_3 = 12.31$, $p = 0.0064$; electronic supplementary material, tables S6 and S7), which was driven exclusively by a fecundity cost of reproduction under control conditions in the highest elevation (3340 m; electronic supplementary material, figure S3). We detected no fecundity cost of failed reproduction (electronic supplementary material, table S6).

## (d) Lifetime costs
The lifetime costs of resource allocation to first-year failed fruits differed as a function of garden elevation by treatment ($\chi^2_4 = 20.96$, $p = 0.0003$; electronic supplementary material, table S8). Lifetime fecundity declined with the proportion of failed fruits across all environments, except in the control treatment at the second-highest elevation (3133 m); the negative effects of high proportional production of failed siliques were most pronounced in low elevations and under snow removal at high elevations (figure 3*a,d,e*; electronic supplementary material, table S9).

The synergistic effect of successful and failed reproduction on lifetime fecundity varied as a function of garden elevation, (successful first-year fitness × first-year failed reproduction ×

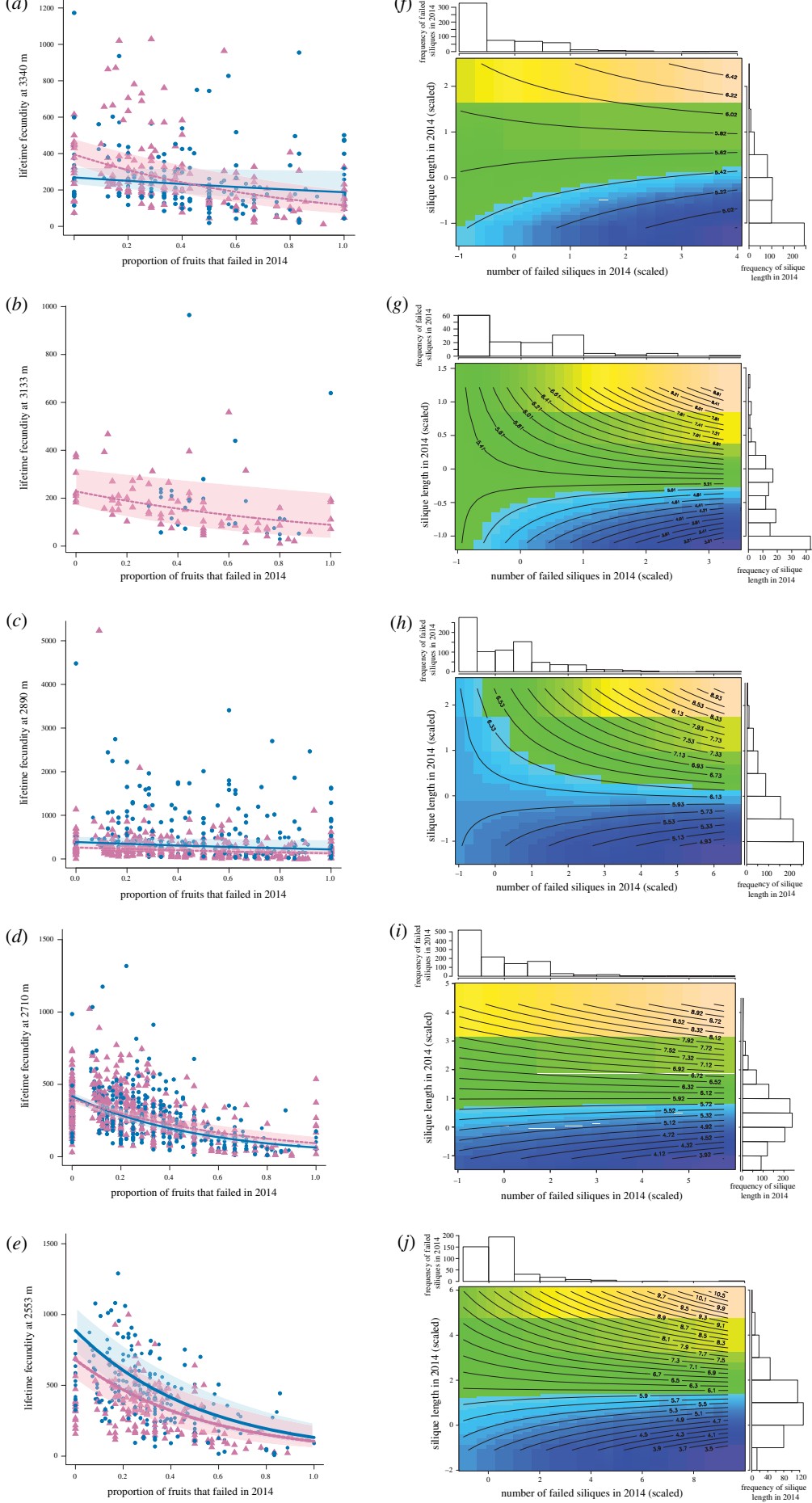

**Figure 3.** (Caption Continued)

**Figure 3.** (*Continued.*) Lifetime fitness among reproductive individuals (cumulative silique length between 2014 and 2019) declined as a function of the proportion of fruits that failed in the first year (*a–e*), and varied in response to the interactive effects of first-year fitness and failed reproduction (*f–j*). Panels (*a–e*) show predicted regression relationships from mixed-effect gamma regression with 95% confidence intervals derived from $n = 2000$ bootstraps in blue for control and pink for snow removal (printed version: solid lines and light grey for control, and dashed lines and dark grey for snow removal). We omitted the non-significant regression line for the control treatment at 3133 m (*b*). The contour plots in (*f–j*) show how first-year successful reproduction (silique length in 2014) and failed reproduction (number of failed siliques in 2014) interacted to influence lifetime fitness at four elevations (*g–j*), but not in the highest elevation (*f*). The adjacent histograms show the frequency distribution of first-year silique length (to the right of each panel) and fruit loss (on top of each panel). (Online version in colour.)

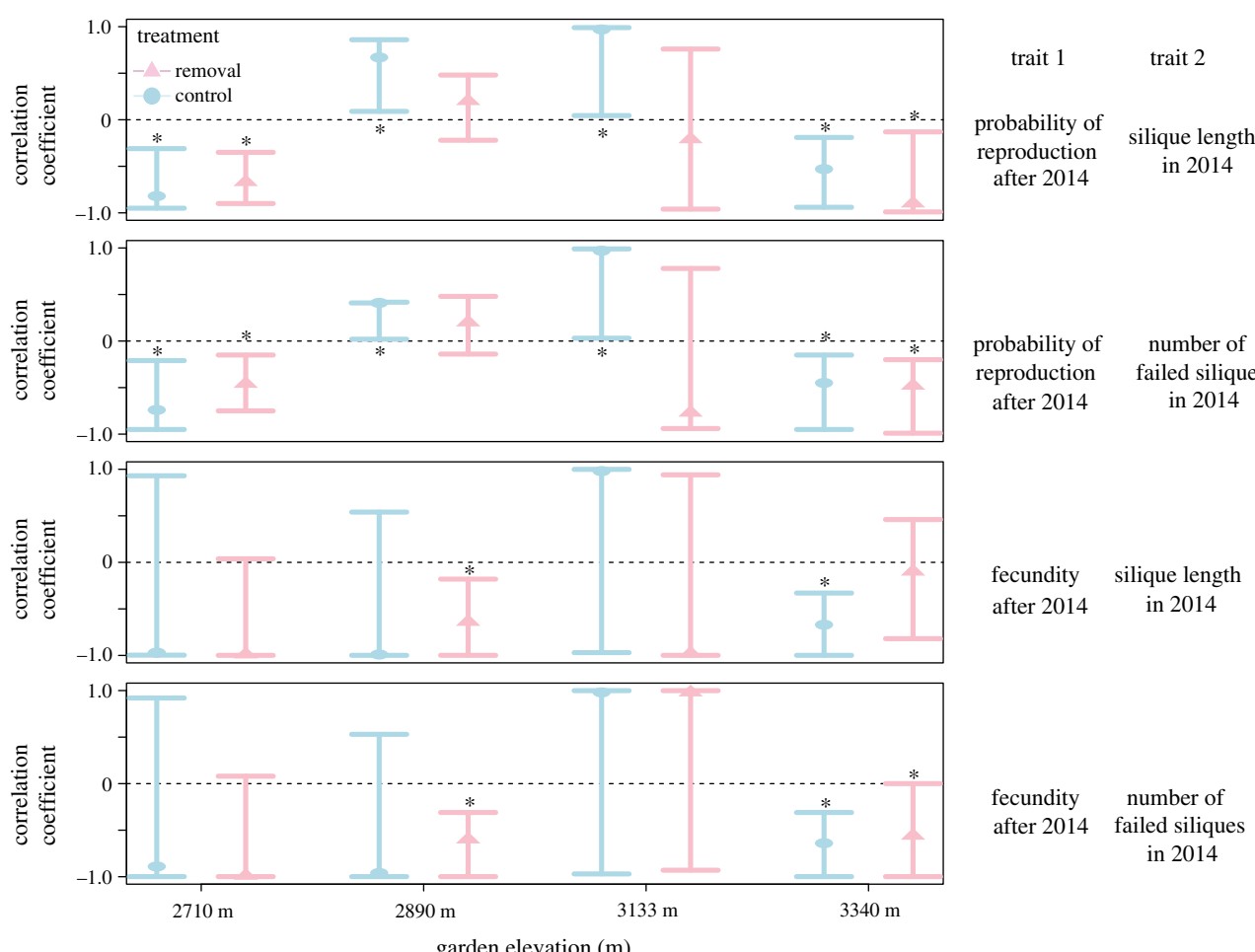

**Figure 4.** Genetic correlations and 95% credible intervals across garden elevation and treatments (control in blue versus snow removal in pink; in printed version for each garden elevation, control is on the left (circles and light grey) and removal is on the right (triangles and dark grey)) for the fitness trade-offs between trait 1 (future reproduction) and trait 2 (first-year fitness or failed reproduction). Asterisks indicate significant correlations. (Online version in colour.)

garden elevation: $\chi^2_4 = 10.40$, $p = 0.03$; figure 3*f–j*; electronic supplementary material, tables S10 and S11). In four of the five elevations, lifetime fecundity peaked under high allocation to both successful and failed initial reproduction, but few individuals occupied that part of the fitness landscape (figure 3*f–j*). For individuals with lower first-year fitness, allocation to failed initial reproduction was detrimental in the lowest elevation gardens (2553, 2710 m). By contrast, the lifetime costs of initial failed reproduction dampened at higher elevation gardens (2890 and 3133 m). We found no significant interaction in the highest elevation.

## (e) Genetic correlations
All patterns that emerged at the phenotypic level were also present in genetic correlations, validating that costs of reproduction detected in phenotypic analyses result from genotypic variation in resource allocation and acquisition. At low and high elevations (2710 and 3340 m), we detected trade-offs between first-year fitness and failed reproduction

for the probability of future reproduction under both control and snow removal (figure 4; electronic supplementary material, table S12). At mid-elevation gardens (2890 and 3133 m), positive genetic correlations indicated that high first-year fitness was significantly associated with high future reproductive odds in control environments (figure 4). We found negative genetic correlations between first-year fitness and future fecundity under snow removal at mid-elevation (2890 m) and under ambient snow dynamics at the highest elevation (3340 m) (figure 4). Finally, failed reproduction incurred a future fecundity cost in ambient conditions at mid-elevation (2890 m) and in both treatments at the highest elevation (3340 m).

## 4. Discussion
Our large-scale field experiment demonstrated that environmental conditions can fundamentally alter the cost of reproduction, a key life-history trade-off that underlies the

evolution of perenniality. Concordant with predictions, costs of reproduction were more apparent in arid, lower elevation sites. For some fitness components, costs of reproduction also emerged under control conditions at our highest elevation. By contrast, in the mid-elevations, high investment in first-year reproductive effort lengthened lifespan (both treatments) and increased future fecundity (control treatment only). Snow removal at these mid-elevation sites eliminated this positive association, generating pronounced fecundity costs of reproduction for one site in genetic correlation analyses. These results suggest that climate change could alter life-history evolution across the elevational range of the species.

The initial investment in reproduction decreased future fitness at the lower elevations (especially at 2710 m) even under ambient conditions. Climate change has increased aridity in this system through reductions in snowpack and accelerations of snowmelt [41,50,58], which could exacerbate costs of reproduction at the lower end of the elevational gradient. Concordant with our results, drought stress imposed a strong growth cost of reproduction in *B. stricta* in a greenhouse experiment [59]. We suggest that these costs will become more apparent as climate change continues to augment drought stress in this landscape. Integral projection models indicate that local *B. stricta* populations are already in decline at low elevations [41], and heightened costs of reproduction could hasten these declines if they reduce lifetime fitness.

By contrast with the costs of reproduction at low elevations, at mid-elevations (2890 and 3133 m), longevity increased with initial investment in reproduction and individuals with high fitness in the first growing season were more likely to reproduce again in control treatments. These mid-elevations experience the mild climates, with greater snowpack, later snowmelt, lower temperatures and reduced evapotranspiration than low elevation site, without the shorter growing seasons of higher elevations (see table S1 in [41]). We propose that the higher resource availability at these mid-elevation sites allows individuals to maintain resource uptake during reproduction, thereby compensating for costs associated with reproduction [16,17]. The positive associations between initial reproductive effort and future fitness could arise if some individuals occur in resource-rich microsites within these gardens. However, owing to significantly positive genetic correlations between initial fitness and the probability of reproduction after 2014 in control treatments at these mid-elevation sites, we propose that genotypic variation in resource acquisition rates underlies these positive associations [16]. By contrast, strong fecundity costs of reproduction emerged in genetic correlation models at 2890 m under snow removal, which increases drought stress [41]. These results suggest that climate change, as simulated by our experimental manipulation of snow dynamics, can generate pronounced costs of reproduction by severely reducing the amount of available resources.

At the highest elevation (3340 m), the probability of future reproduction declined with initial reproduction in both treatments. Phenotypic and genetic models also revealed fecundity costs of reproduction under control conditions but not snow removal. As this site has the highest water availability along this elevation gradient [41], the costs of reproduction are probably driven by short growing seasons rather than aridity. The restricted growing season limits the timeframe for resource acquisition, which could augment the costs of reproduction. Indeed Sletvold & Ågren [14] documented more intense life-history trade-offs under shorter growing season for the orchid *Dactylorhiza lapponica*. Snow removal extended the growing season by an average of 9.8 days at this elevation [41]. If the main effect of climate change at high elevations is to lengthen the growing season and dampen costs of reproduction, population growth rates could expand, concordant with projections from a complementary experiment [41].

## (a) Costs of failed reproduction

Plants regularly produce flowers and fruits that fail. Selective fruit abortion could enable plants to resorb and reallocate resources to alternative functions, or resources allocated to failed fruits could be lost, intensifying costs of reproduction [12]. We found that high initial fruit loss reduced plant longevity and depressed the probability of subsequent reproduction at lower elevations. Lifetime fecundity declined with increasing proportional allocation to failed fruits in the first growing season across elevations (except in the control at 3133 m where we found no pattern). Moreover, lifetime fecundity costs of failed fruit production were especially apparent for individuals with low initial successful reproduction (figure 3). Thus, fruit loss generally reduces future and lifetime fitness, and in most cases seems to be a maladaptive response to stress and resource limitation.

Given the costs of reproductive failure, why would plants produce surplus reproductive structures? Proposed hypotheses include: uncertainties in pollination success; predation of flowers, fruits and seeds; the unpredictable nature of future resource availability; and the potential to improve seed quality by the selective abscission of low-quality fruits [33]. Although lifetime fecundity declined with early reproductive failure across elevations, a benefit of aborting fruits emerged at mid-elevations. High initial failed reproduction was associated with increased longevity in both treatments in phenotypic models, and with higher future reproductive odds in control treatments at mid-elevations, as indicated by the genetic correlation models. Additionally, in most environments, lifetime fitness peaks were set by the few individuals that produced high numbers of matured and failed fruits during the first year, indicating high-fitness individuals may benefit from fruit abortion. Under benign conditions and high resource availability, the increase in reproductive success owing to the overproduction of flowers and fruits could outweigh the additional cost associated with wasting resources [16]. Additionally, under these conditions, plants may be able to moderate costs associated with fruit loss via continuous resource uptake, and could also be more efficient at nutrient resorption from aborted fruits. Thus, for high-fitness individuals under benign conditions, fruit abortion may be an adaptive strategy to adjust allocation to the current and future reproductive effort.

## (b) Life-history evolution under climate change

Climate change influences vital rates that shape life-history strategies [60–62]. It follows that the costs and benefits of alternative life-history strategies could also shift under climate change, yet little is known about life-history evolution in novel environmental conditions [28,39]. Life-history evolution is strongly driven by stage-specific mortality rates [40]. For example, individuals from low elevation *Erysimun capitatum* (Brassicaeae) populations had higher mortality at all life stages compared to those from higher elevations, and low elevation plants

reproduced more quickly and were more frequently semelparous than alpine plants [39]. As adult survival is a key demographic parameter maintaining alpine populations [38], long-lived, iteroparous species should allocate resources to reproduction in a way that limits survival costs [63]. Yet, our climate change manipulations shifted the balance from a benefit of reproduction in our mid-elevation sites to a cost of reproduction, with reduced future fecundity after high initial allocation to reproduction in stressful conditions. By contrast, in our highest elevation site, snow removal dampened the severe costs of reproduction observed under ambient conditions.

As climate envelopes move upslope in mountainous regions and toward the poles latitudinally, local populations may no longer perform optimally in new environmental conditions at their home sites [41,64,65]. If the benefits of a slower life strategy decline in mid-elevation alpine habitats, long-lived iteroparous species may transition towards shorter lifespans under increasingly stressful climates [39,40]. Similarly, at lower elevations, high adult mortality after reproduction may ultimately favour annual or biennial life strategies. Intensified costs of reproduction under climate change could disrupt the evolution of life-history strategies and shorten plant lifespans under novel abiotic stress.

## 5. Conclusion

In our long-term experimental manipulation, we demonstrate that climate change can shift costs of reproduction in a perennial subalpine forb. Costs of reproduction were prevalent in arid, lower elevations and under the short growing seasons at the highest elevation. Additionally, pronounced costs of reproduction emerged under climate change simulations at mid-elevations. We hypothesize that shifts in costs of reproduction could alter life-history evolution, potentially reducing longevity in all but the highest elevation locales. We encourage future studies to examine how shifts in costs of reproduction under climate change will affect plant population dynamics.

Data accessibility. All data and R scripts are available from the Dryad Digital Repository: https://doi.org/10.5061/dryad.1zcrjdfrf [66].

Authors' contributions. E.H. analysed data and wrote the manuscript. J.T.A. acquired funding, collected data, analysed data and wrote the manuscript. S.M.W. collected data and edited the manuscript. All authors worked on the figures.

Competing interests. We declare we have no competing interests.

Funding. Funding came from the National Science Foundation (grant no. DEB-1553408).

Acknowledgements. We thank Megan Peterson, Inam Jameel, Derek Denney, Rachel MacTavish, Mia Rochford, Sam Day and Tom Pendergast for discussions. Jennie Reithel provided logistical support and facilitated permits, and the Crested Butte Land Trust and Estess family allowed us to establish gardens. We are grateful to field assistants and backcountry skiers, including Bashira Chowdhury, Caroline Daws, Anna Battiata, Caroline Boerner, Tyler Morrison, Sahana Srivatsan, Noah Workman, Hayley Nagle, Dylan Proudfoot, Turner Kilgore, Cameron Smith, Austen Beason, Ben Ammon, Alex Tiberio, Peter Innes, Kristi Haner, Andres Esparza, Dustin Eldridge, Hunter Grimess, Erik Cheslock, Evan Ross, Jill Gall, Megan Verner-Crist, Margaret Manto and Curtis Beutler.

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
