## [Peer Review File · Proceedings of the Royal Society B: Biological Sciences]

Review History

RSPB-2020-3134.R0 (Original submission)

Review form: Reviewer 1

Recommendation

Major revision is needed (please make suggestions in comments)

Scientific importance: Is the manuscript an original and important contribution to its field?

Good

General interest: Is the paper of sufficient general interest?

Excellent

Quality of the paper: Is the overall quality of the paper suitable?

Good

Is the length of the paper justified?

Yes

Should the paper be seen by a specialist statistical reviewer?

No

Do you have any concerns about statistical analyses in this paper? If so, please specify them explicitly in your report.

No

It is a condition of publication that authors make their supporting data, code and materials available - either as supplementary material or hosted in an external repository. Please rate, if applicable, the supporting data on the following criteria.

Is it accessible?

Yes

Is it clear?

Yes

Is it adequate?

Yes

Do you have any ethical concerns with this paper?

No

Comments to the Author

This study examines how costs of reproduction varies with experimental manipulations of snow cover at sites different altitudes in a perennial herb. The experiment is very ambitious and the data collected appropriate. No doubt, this material could form the basis for a valuable contribution. My main concerns with the current version of manuscript, that I detail below, is that the theoretical background and logic of the analyses is unclear.

The provided theoretical framework is insufficient to understand the logic behind the analyses and the interpretation of the results. For example, the second paragraph confuses factors that influence the expression of costs, such as the availability of resources and abiotic conditions, with factors that influence or ability to detect costs, such as the variation in resource acquisition vs. variation in resource allocation. I think that this distinction is crucial and I also think that a clearer theoretical background in this respect would improve the interpretation of the results and provide a motivation for the study design (see below). One example of where such a clearer theoretical framework would be useful is in the interpretation of what the presence vs. absence of significant correlations in response to treatments (e.g. lines 315-318). Now such differences are interpreted as novel costs generated by climate change, but it is also possible that treatments alter the relative size of variation in resource allocation to variation in resource acquisition.

I also lack some mentioning of how the problems associated with a large variation in resource acquisition vs. the variation in resource allocation potentially can be dealt with. The most frequently used approach to deal with this problem is to use experimental manipulations of reproductive effort. Another possibility is to focus in genetic correlations rather than phenotypic correlations. Later, the former approach is dismissed in the current study while the latter approach is adopted. I think that the simultaneous assessment of phenotypic and genotypic correlations is actually one of the major strengths of the current study. I would therefore like to see the results of the genetic correlations presented in more detail and more explicitly compared with the results of the phenotypic correlations. I would also like to see a better motivation for why assessing also genetic correlations is important in the introduction, and how results from phenotypic vs. genotypic correlations should be interpreted.

A relatively large part of the paper is concerned with the role of reproductive failure. I am not saying that this is not an interesting aspect, but have a problem to understand from the text in the introduction why this is a particularly interesting aspect that differ qualitatively from other aspects of variation in reproductive effort and fitness. Fruit abortion has often been treated as one of several steps in which plants adjust reproductive effort to available resources (e.g. Lloyd 1980,

New Phytol. 86:69-79). The lack of a thorough theoretical background and clear definition of concepts and terms also sometimes results in statements that are difficult to interpret, e.g. “The synergistic effects of successful and failed reproduction on lifetime fitness ..” (line 267). If the authors want to keep the focus on reproductive failure and fruit abortion, I think that the logic behind the analyses where effects of reproductive effort and reproductive failure is are explored simultaneously, and why reproductive failure is not considered a component of fitness, needs to be much better explained.

In several places, the effects of treatments are referred to as effects of climate change (e.g. on lines 102 and 149). I do not think that this is correct, and that the authors need to clearly distinguish between effects of treatments and possible implications of these in a climate change context. Moreover, the references to effects of climate change do also sometimes appear to be a bit poorly motivated, e.g. on lines 63-64.

The results section is a bit hard to follow as it is not clear how the results relate to the main predictions, and would suggest the authors to focus more explicitly on the tests of the predictions in the result section. It is also a bit difficult to grasp all the significant two- and three-way interactions, some of which seem hard to find a good biological explanation for (e.g. lines 228-237). The structure of the results and discussion sections should also be better aligned.

Specific comments:

Lines 29-30, and in the main text: Perhaps this is a matter of taste, but I would not say that life history theory posits a cost of reproduction, but rather that the presence of costs is an empirically based pattern. Life history theory does, however, posit how this cost will influence life history evolution in different environments.

Line 31, and in the main text: Clarify how trade-offs are altered - through changes in costs or benefits?

Line 37, and in the main text: What do you mean by “novel” cost?

Lines 76-77: I don’t think that this statement is correct, and that it leads to an overstatement of the novelty of the current study in this respect.

Line 83: Unclear to me why fruit abortion should exaggerate costs.

Line 83-85: The meaning of this sentence is unclear to me. Whether fruit abortion is associated with a strategy or not would hardly influence the costs!?

Lines 92-93: I don’t consider this to be a valid argument for why experimental manipulations were not appropriate in this study. As long as manipulations alter reproductive effort, they could still be useful.

Lines 101-103: I think it would be better if you explained what effects of treatments you expected and how you expected these treatments to interact with site.

Lines 110-111: I would argue the other way; if fecundity depends on pollinator availability and is more variable, then this would result in larger variation in resource allocation, and in increasing possibilities to detect costs.

Line 150: I am not familiar with the concept of “failed reproductive effort”. As reproductive effort is often used to describe the fraction of available resources that are used for reproduction, the concept does not really make sense to me. Reproduction can fail but not reproductive effort.

Review form: Reviewer 2

Recommendation

Accept with minor revision (please list in comments)

Scientific importance: Is the manuscript an original and important contribution to its field?

Excellent

General interest: Is the paper of sufficient general interest?

Excellent

Quality of the paper: Is the overall quality of the paper suitable?

Good

Is the length of the paper justified?

Yes

Should the paper be seen by a specialist statistical reviewer?

No

Do you have any concerns about statistical analyses in this paper? If so, please specify them explicitly in your report.

No

It is a condition of publication that authors make their supporting data, code and materials available - either as supplementary material or hosted in an external repository. Please rate, if applicable, the supporting data on the following criteria.

Is it accessible?

N/A

Is it clear?

N/A

Is it adequate?

N/A

Do you have any ethical concerns with this paper?

No

Comments to the Author

The article "Costs of reproduction under experimental climate change across elevations in the perennial forb *Boechera stricta*" by Hamann, Wadgymar, and Anderson is an ambitious common garden experiment conducted over six years at five locations along an altitudinal gradient in the US Rocky Mountains. It included snow removal treatments at all gardens, which is an impressive undertaking, given these altitudes and the length of time involved. The results are an important contribution to our understanding of how climate change and drought impact reproduction in a perennial montane/alpine plant species. However, I felt there were some areas of the manuscript that needed further clarification or detail. The Introduction lacked specifics about introducing the measures used in this study, while the Discussion introduced concepts related to the evolution of perennality, give trade-offs to reproduction. More details of the specific types of costs should be highlighted in the Intro, as well as their potential role in shaping evolution.

Introduction

Minor comments:

Line 55: Introduce the word iteroparous, because it is used later without having been contrasted with the use of semelparous in Line 54.

Line 85: What is meant by bet-hedging? I can guess, but further explanation will help those readers less familiar with allocation/trade-off literature.

Major comments:

The costs of reproduction referenced in other studies should introduce your own approach for quantifying costs (2nd and 3rd paragraphs). I found the Introduction lacked specifics about these costs, and was at times repetitive. For example, the sentences that start on lines 57 and 59 essentially make similar points. Furthermore, the 2nd and 3rd paragraph of the Intro were more or less the same. In both cases, there were references indicating that costs of reproduction increased under some scenarios, but these costs were not defined and explained. Any indication how these costs impacted future growth or reproduction? How do these relate to your study? You look at future fecundity and survival, so please establish the motivation for your measurements.

In the description of tests, on Line 98, it is predicted that "novel" costs of reproduction emerge. What do you mean by "novel?" Please contrast with previous studies (this ties in with the comments above). Also, this will help prepare readers the methods, where you introduce "longevity costs," "fecundity costs," etc.

Methods

Minor comments:

Line 120: How many blocks per garden?

Line 126: It took me a bit to understand that these supplementary tables are from "reference" 40. Please clarify. Also, how do the six years of your study compare with historical averages of temperature and snowfall? An important aspect of long-term studies like this is that it can encompass a range of variation across years.

Line 127: Plant diameter. Do you mean rosette diameter? Stem diameter?

Line 155: Instead of using "garden" as a variable, how about referring to it as "altitude?"

Line 163: "Geographic distance". Is this again the difference between "source" and "elevation?"

Line 165-166: Longevity costs. Please define what you mean by longevity costs. Also, in the caption for Figure 1, you describe what ratios that are $>$ or $<$ 1 indicate. Please describe here, as well.

Major comments:

Fitness and fecundity aren't always clearly distinguished. I think the intent was to use fitness for 2014, and then fecundity for the rest of the experiment, but this was not always clear. If I'm correct about that interpretation, I think specifying "first-year fitness" and "lifetime fecundity" will help.

Additionally, the section on "lifetime costs" starting on line 188 appeared similar to the section on "fecundity costs" on the previous page. An introductory sentence for each paragraph should indicate what each test reveals. (The first sentence on lines 172-173 doesn't tell me).

Results:

The paragraph starting on Line 233 is not necessary in its entirety, since it more or less reports the same results as the previous paragraph.

Line 242-243: When you state "significant cost of reproduction," indicate that you mean a cost in terms of future reproduction, since you are examining multiple different "costs".

Line 247: By "environmental context," do you mean elevation?

Line 267: "The synergistic effect of successful and failed reproductive effect..." This sentence was unclear.

Discussion:

Line 288: Again "environmental context" should be clarified. Elevation? Precipitation? Also, I don't think you've previously discussed the evolution of perenniality, so that statement isn't the best way to introduce the discussion. This comes up again later in the Discussion (last page), and

should be addressed more in the Introduction.

Line 311: "Reduced evapotranspiration." Do you have data for this? Or was this from a prior publication?

Line 312: Do you mean "shorter" rather "shortened"?

Line 315: Expand on the hypothesis regarding variation in resource acquisition. How is this helping plants more at some elevations vs. others?

I like introducing the concept of how the variation of these trade-offs over elevations may shape evolution, but some context for this should be included in the Introduction.

Decision letter (RSPB-2020-3134.R0)

15-Feb-2021

Dear Dr Anderson:

Your manuscript has now been peer reviewed and the reviews have been assessed by an Associate Editor. All of us are impressed by the scope of your study and its potential to inform the current discussion surrounding climate change. However, both reviewers also made a number of thoughtful comments on aspects of your manuscript that need to be addressed before we can further consider it for publication. I also second the AE that while the reviewers' comments are fairly straightforward, I encourage you to think about the larger picture as you re-frame your manuscript in order to maximize your impact. Their reviews (not including confidential comments to the Editor) and the comments from the Associate Editor are included at the end of this email for your reference. I invite you to revise your manuscript to address them.

Research ethics:

Use of animals and field studies:

It is a condition of publication that you make available the data and research materials supporting the results in the article. Please see our Data Sharing Policies (<https://royalsociety.org/journals/authors/author-guidelines/#data>). Datasets should be deposited in an appropriate publicly available repository and details of the associated accession number, link or DOI to the datasets must be included in the Data Accessibility section of the article (<https://royalsociety.org/journals/ethics-policies/data-sharing-mining/>). Reference(s) to datasets should also be included in the reference list of the article with DOIs (where available).

Please submit a copy of your revised paper within three weeks. If we do not hear from you within this time your manuscript will be rejected. If you are unable to meet this deadline please let us know as soon as possible, as we may be able to grant a short extension.

Best wishes,

Dr Sarah Brosnan
 Editor, Proceedings B
 mailto: proceedingsb@royalsociety.org

Associate Editor

Comments to Author:

I have now received two reviews of the manuscript, “Costs of reproduction under experimental climate change across elevations in the perennial forb *Boechera stricta*” The manuscript reports the results of a multiyear, multisite study to measure cost of reproduction in an herbaceous high-elevation perennial under control and early-warming conditions. The work found that low and high elevation environments were most stressful and yielded the greatest cost of reproduction. The work is important in evaluating cost of reproduction under natural and manipulated (i.e. stress) conditions over much of the environmental breadth of a species, informing our understanding of the conditions in which costs are expressed and how they may be enhanced under climate change.

The reviews are thorough, thoughtful, and express somewhat similar themes. Overall, they were quite positive about the design and scope of the project, the analyses and the results. They did have comments about the framing and motivation of aspects of the work, as well as word usage in some areas. These are aspects of the manuscript that could be addressed via revision. However, given that some of the comments overlap between reviewers, I encourage the authors to think about the broader issues the reviewers raise and take the comments seriously.

Comments from the Associate Editor:

Fig 3. The legend wasn't much help in interpreting Figure 3 F-J.

Line 346. This line seems to ascribe a causal arrow to a genetic correlation. The causal arrow is your interpretation – be a bit more circumspect.

Reviewer(s)' Comments to Author:

Referee: 1

Comments to the Author(s)

This study examines how costs of reproduction varies with experimental manipulations of snow cover at sites different altitudes in a perennial herb. The experiment is very ambitious and the data collected appropriate. No doubt, this material could form the basis for a valuable contribution. My main concerns with the current version of manuscript, that I detail below, is that the theoretical background and logic of the analyses is unclear.

The provided theoretical framework is insufficient to understand the logic behind the analyses and the interpretation of the results. For example, the second paragraph confuses factors that influence the expression of costs, such as the availability of resources and abiotic conditions, with factors that influence or ability to detect costs, such as the variation in resource acquisition vs. variation in resource allocation. I think that this distinction is crucial and I also think that a clearer theoretical background in this respect would improve the interpretation of the results and provide a motivation for the study design (see below). One example of where such a clearer theoretical framework would be useful is in the interpretation of what the presence vs. absence of significant correlations in response to treatments (e.g. lines 315-318). Now such differences are interpreted as novel costs generated by climate change, but it is also possible that treatments alter the relative size of variation in resource allocation to variation in resource acquisition.

I also lack some mentioning of how the problems associated with a large variation in resource acquisition vs. the variation in resource allocation potentially can be dealt with. The most frequently used approach to deal with this problem is to use experimental manipulations of reproductive effort. Another possibility is to focus in genetic correlations rather than phenotypic

correlations. Later, the former approach is dismissed in the current study while the latter approach is adopted. I think that the simultaneous assessment of phenotypic and genotypic correlations is actually one of the major strengths of the current study. I would therefore like to see the results of the genetic correlations presented in more detail and more explicitly compared with the results of the phenotypic correlations. I would also like to see a better motivation for why assessing also genetic correlations is important in the introduction, and how results from phenotypic vs. genotypic correlations should be interpreted.

A relatively large part of the paper is concerned with the role of reproductive failure. I am not saying that this is not an interesting aspect, but have a problem to understand from the text in the introduction why this is a particularly interesting aspect that differ qualitatively from other aspects of variation in reproductive effort and fitness. Fruit abortion has often been treated as one of several steps in which plants adjust reproductive effort to available resources (e.g. Lloyd 1980, *New Phytol.* 86:69-79). The lack of a thorough theoretical background and clear definition of concepts and terms also sometimes results in statements that are difficult to interpret, e.g. "The synergistic effects of successful and failed reproduction on lifetime fitness .." (line 267). If the authors want to keep the focus on reproductive failure and fruit abortion, I think that the logic behind the analyses where effects of reproductive effort and reproductive failure is are explored simultaneously, and why reproductive failure is not considered a component of fitness, needs to be much better explained.

In several places, the effects of treatments are referred to as effects of climate change (e.g. on lines 102 and 149). I do not think that this is correct, and that the authors need to clearly distinguish between effects of treatments and possible implications of these in a climate change context. Moreover, the references to effects of climate change do also sometimes appear to be a bit poorly motivated, e.g. on lines 63-64.

The results section is a bit hard to follow as it is not clear how the results relate to the main predictions, and would suggest the authors to focus more explicitly on the tests of the predictions in the result section. It is also a bit difficult to grasp all the significant two- and three-way interactions, some of which seem hard to find a good biological explanation for (e.g. lines 228-237). The structure of the results and discussion sections should also be better aligned.

Specific comments:

Lines 29-30, and in the main text: Perhaps this is a matter of taste, but I would not say that life history theory posits a cost of reproduction, but rather that the presence of costs is an empirically based pattern. Life history theory does, however, posit how this cost will influence life history evolution in different environments.

Line 31, and in the main text: Clarify how trade-offs are altered - through changes in costs or benefits?

Line 37, and in the main text: What do you mean by "novel" cost?

Lines 76-77: I don't think that this statement is correct, and that it leads to an overstatement of the novelty of the current study in this respect.

Line 83: Unclear to me why fruit abortion should exaggerate costs.

Line 83-85: The meaning of this sentence is unclear to me. Whether fruit abortion is associated with a strategy or not would hardly influence the costs!?

Lines 92-93: I don't consider this to be a valid argument for why experimental manipulations were not appropriate in this study. As long as manipulations alter reproductive effort, they could still be useful.

Lines 101-103: I think it would be better if you explained what effects of treatments you expected and how you expected these treatments to interact with site.

Lines 110-111: I would argue the other way; if fecundity depends on pollinator availability and is more variable, then this would result in larger variation in resource allocation, and in increasing possibilities to detect costs.

Line 150: I am not familiar with the concept of “failed reproductive effort”. As reproductive effort is often used to describe the fraction of available resources that are used for reproduction, the concept does not really make sense to me. Reproduction can fail but not reproductive effort.

Referee: 2

Comments to the Author(s)

The article “Costs of reproduction under experimental climate change across elevations in the perennial forb *Boechera stricta*” by Hamann, Wadgymar, and Anderson is an ambitious common garden experiment conducted over six years at five locations along an altitudinal gradient in the US Rocky Mountains. It included snow removal treatments at all gardens, which is an impressive undertaking, given these altitudes and the length of time involved. The results are an important contribution to our understanding of how climate change and drought impact reproduction in a perennial montane/alpine plant species. However, I felt there were some areas of the manuscript that needed further clarification or detail. The Introduction lacked specifics about introducing the measures used in this study, while the Discussion introduced concepts related to the evolution of perenniality, give trade-offs to reproduction. More details of the specific types of costs should be highlighted in the Intro, as well as their potential role in shaping evolution.

Introduction

Minor comments:

Line 55: Introduce the word iteroparous, because it is used later without having been contrasted with the use of semelparous in Line 54.

Line 85: What is meant by bet-hedging? I can guess, but further explanation will help those readers less familiar with allocation/trade-off literature.

Major comments:

The costs of reproduction referenced in other studies should introduce your own approach for quantifying costs (2nd and 3rd paragraphs). I found the Introduction lacked specifics about these costs, and was at times repetitive. For example, the sentences that start on lines 57 and 59 essentially make similar points. Furthermore, the 2nd and 3rd paragraph of the Intro were more or less the same. In both cases, there were references indicating that costs of reproduction increased under some scenarios, but these costs were not defined and explained. Any indication how these costs impacted future growth or reproduction? How do these relate to your study? You look at future fecundity and survival, so please establish the motivation for your measurements.

In the description of tests, on Line 98, it is predicted that “novel” costs of reproduction emerge. What do you mean by “novel?” Please contrast with previous studies (this ties in with the comments above). Also, this will help prepare readers the methods, where you introduce “longevity costs,” “fecundity costs,” etc.

Methods

Minor comments:

Line 120: How many blocks per garden?

Line 126: It took me a bit to understand that these supplementary tables are from “reference” 40. Please clarify. Also, how do the six years of your study compare with historical averages of

temperature and snowfall? An important aspect of long-term studies like this is that it can encompass a range of variation across years.

Line 127: Plant diameter. Do you mean rosette diameter? Stem diameter?

Line 155: Instead of using "garden" as a variable, how about referring to it as "altitude?"

Line 163: "Geographic distance". Is this again the difference between "source" and "elevation?"

Line 165-166: Longevity costs. Please define what you mean by longevity costs. Also, in the caption for Figure 1, you describe what ratios that are $>$ or $<$ 1 indicate. Please describe here, as well.

Major comments:

Fitness and fecundity aren't always clearly distinguished. I think the intent was to use fitness for 2014, and then fecundity for the rest of the experiment, but this was not always clear. If I'm correct about that interpretation, I think specifying "first-year fitness" and "lifetime fecundity" will help.

Additionally, the section on "lifetime costs" starting on line 188 appeared similar to the section on "fecundity costs" on the previous page. An introductory sentence for each paragraph should indicate what each test reveals. (The first sentence on lines 172-173 doesn't tell me).

Results:

The paragraph starting on Line 233 is not necessary in its entirety, since it more or less reports the same results as the previous paragraph.

Line 242-243: When you state "significant cost of reproduction," indicate that you mean a cost in terms of future reproduction, since you are examining multiple different "costs".

Line 247: By "environmental context," do you mean elevation?

Line 267: "The synergistic effect of successful and failed reproductive effect..." This sentence was unclear.

Discussion:

Line 288: Again "environmental context" should be clarified. Elevation? Precipitation? Also, I don't think you've previously discussed the evolution of perenniality, so that statement isn't the best way to introduce the discussion. This comes up again later in the Discussion (last page), and should be addressed more in the Introduction.

Line 311: "Reduced evapotranspiration." Do you have data for this? Or was this from a prior publication?

Line 312: Do you mean "shorter" rather "shortened"?

Line 315: Expand on the hypothesis regarding variation in resource acquisition. How is this helping plants more at some elevations vs. others?

I like introducing the concept of how the variation of these trade-offs over elevations may shape evolution, but some context for this should be included in the Introduction.

Author's Response to Decision Letter for (RSPB-2020-3134.R0)

See Appendix A.

Decision letter (RSPB-2020-3134.R1)

17-Mar-2021

Dear Dr Anderson

I am pleased to inform you that your manuscript entitled "Costs of reproduction under experimental climate change across elevations in the perennial forb *Boechera stricta*" has been accepted for publication in Proceedings B.

Data Accessibility section

Open Access

Paper charges

Sincerely,

Dr Sarah Brosnan

Associate Editor:

Board Member

Comments to Author:

Thank you for your thorough and thoughtful revision of "Costs of reproduction under experimental climate change across elevations in the perennial forb *Boechera stricta*." I appreciate

the many changes that you made in response to the reviewer's comments, in particular the substantial edits to the introduction. With these changes, I think this manuscript will make a strong addition to Proceedings B.

Appendix A

Dear Dr. Brosnan,

Thank you for the opportunity to resubmit our manuscript "Costs of reproduction under experimental climate change across elevations in the perennial forb *Boecheera stricta*" (RSPB-2020-3134). We are grateful for the thoughtful critiques of the reviewers, and have revised the manuscript thoroughly. In particular, we clarified the framework of the study, our predictions, the terminology used to refer to the examined fitness components, and how the identified shifts in costs of reproduction under climate change may shape the evolution of life history strategies.

Below, we describe our revisions in point-by-point responses (in blue text) to the comments of each reviewer. We have provided one clean version of the revised manuscript along with one version showing all revisions in Track Changes. The line numbers we reference in this response letter refer to both versions.

Sincerely,
Elena Hamann, Susana Wadgymar, and Jill Anderson

Associate Editor

Comments to Author:

I have now received two reviews of the manuscript, "Costs of reproduction under experimental climate change across elevations in the perennial forb *Boecheera stricta*". The manuscript reports the results of a multiyear, multisite study to measure cost of reproduction in an herbaceous high-elevation perennial under control and early-warming conditions. The work found that low and high elevation environments were most stressful and yielded the greatest cost of reproduction. The work is important in evaluating cost of reproduction under natural and manipulated (i.e. stress) conditions over much of the environmental breadth of a species, informing our understanding of the conditions in which costs are expressed and how they may be enhanced under climate change.

The reviews are thorough, thoughtful, and express somewhat similar themes. Overall, they were quite positive about the design and scope of the project, the analyses and the results. They did have comments about the framing and motivation of aspects of the work, as well as word usage in some areas. These are aspects of the manuscript that could be addressed via revision. However, given that some of the comments overlap between reviewers, I encourage the authors to think about the broader issues the reviewers raise and take the comments seriously.

Dear Associate Editor,

Thank you for the opportunity to revise our manuscript. Your directions along with the comments and suggestions provided by the two reviewers were very valuable, and helped us improve this paper. We thoroughly revised the entire manuscript. In particular, we clarified the framework and motivations in the introduction, and focused on improving our terminology

definitions and predictions. We appreciate that both reviewers raised similar points, which helped us identify areas that lacked precision in our manuscript.

Comments from the Associate Editor:

Fig 3. The legend wasn't much help in interpreting Figure 3 F-J.

Response: Thank you for pointing this out. We recognize that more detail was needed to describe these contour plots and their associated histograms. We added more information to help with the interpretation of Fig. 3 F-J.

Line 346. This line seems to ascribe a causal arrow to a genetic correlation. The causal arrow is your interpretation – be a bit more circumspect.

Response: We have reformulated here (L. 396-398) and elsewhere to avoid overstating our results. Nevertheless, as per the definition in Reznick, 1985, "genetic correlations directly assess the causal relationships between two traits".

Reviewer(s)' Comments to Author:

Referee: 1

Comments to the Author(s)

This study examines how costs of reproduction varies with experimental manipulations of snow cover at sites different altitudes in a perennial herb. The experiment is very ambitious and the data collected appropriate. No doubt, this material could form the basis for a valuable contribution. My main concerns with the current version of manuscript, that I detail below, is that the theoretical background and logic of the analyses is unclear.

Response: We thank Referee # 1 for carefully reading our manuscript, for the overall positive assessment of our work, and for their constructive criticism. We understand that certain aspects of the paper lacked clarity and needed reframing. To address these issues, we thoroughly revised the main manuscript. We address each comment in more detail below.

The provided theoretical framework is insufficient to understand the logic behind the analyses and the interpretation of the results. For example, the second paragraph confuses factors that influence the expression of costs, such as the availability of resources and abiotic conditions, with factors that influence or ability to detect costs, such as the variation in resource acquisition vs. variation in resource allocation. I think that this distinction is crucial and I also think that a clearer theoretical background in this respect would improve the interpretation of the results and provide a motivation for the study design (see below). One example of where such a clearer theoretical framework would be useful is in the interpretation of what the presence vs. absence of significant correlations in response to treatments (e.g. lines 315-318). Now such differences are interpreted as novel costs generated by climate change, but it is also possible that treatments alter the relative size of variation in resource allocation to variation in resource acquisition.

Response: Thank you for this thoughtful comment. We recognize the theoretical background needed clarification and we carefully revised the entire introduction to better delineate the framework of our study. Additionally, we agree that a distinction between factors

that influence the expression of costs or the ability to detect such costs was necessary to better justify our experimental design and allow for a clearer understanding of our results. We revised the introduction to clarify these concepts and the study framework. Specifically, we expanded the second paragraph of the introduction to better distinguish between resource allocation and acquisition (L. 54-74). Additionally, we briefly outline our experimental and analytical approaches to account for microenvironmental variation in resource availability, which as you rightly pointed out below, also allows us to better emphasize our combined approach of examining phenotypic and genetic correlations between initial and future reproduction (L. 109-118).

We also revised the section of the discussion you mention to highlight that reduced resource availability in our snow removal treatment could generate costs of reproduction at mid-elevation locales that we do not see in the control treatment (L. 355-362).

I also lack some mentioning of how the problems associated with a large variation in resource acquisition vs. the variation in resource allocation potentially can be dealt with. The most frequently used approach to deal with this problem is to use experimental manipulations of reproductive effort. Another possibility is to focus in genetic correlations rather than phenotypic correlations. Later, the former approach is dismissed in the current study while the latter approach is adopted. I think that the simultaneous assessment of phenotypic and genotypic correlations is actually one of the major strengths of the current study. I would therefore like to see the results of the genetic correlations presented in more detail and more explicitly compared with the results of the phenotypic correlations. I would also like to see a better motivation for why assessing also genetic correlations is important in the introduction, and how results from phenotypic vs. genotypic correlations should be interpreted.

Response: Again, thank you for raising these issues. We appreciate that our experimental and analytical approaches needed to be better introduced and now highlight our complementary approach of assessing phenotypic and genotypic correlations. As mentioned above, we revised the introduction to explain the use of genetic correlations in combination with phenotypic correlations (L. 109-118). Additionally, we included more information about the genetic correlation models in the methods (L. 240-241). Positive associations between current reproduction and future fitness could arise as an artifact of microenvironmental variation within each common garden, if some individuals occur in resource-rich microsites within these gardens. However, significantly positive genetic correlations indicate that these positive associations do not occur simply because of microenvironmental variation, but through mechanisms such as genotypic variation in resource acquisition rates [1]. We highlight these points in the results (L. 317-319) and discussion section (L. 355-367).

A relatively large part of the paper is concerned with the role of reproductive failure. I am not saying that this is not an interesting aspect, but have a problem to understand from the text in the introduction why this is a particularly interesting aspect that differ qualitatively from other aspects of variation in reproductive effort and fitness. Fruit abortion has often been treated as one of several steps in which plants adjust reproductive effort to available resources (e.g. Lloyd 1980, *New Phytol.* 86:69-79). The lack of a thorough theoretical background and clear definition of concepts and terms also sometimes results in statements that are difficult to interpret, e.g. "The synergistic effects of successful and failed reproduction on lifetime fitness .." (line 267). If the authors want to keep the focus on reproductive failure and fruit abortion, I think that the logic

behind the analyses where effects of reproductive effort and reproductive failure are explored simultaneously, and why reproductive failure is not considered a component of fitness, needs to be much better explained.

Response: We recognize the need to better define our terminology and to clarify how failed reproduction may influence costs of reproduction differently than successful reproduction. Indeed, fruit abortion is generally treated as a strategy to adjust reproductive effort to available resources, but to our knowledge this has rarely been tested empirically. Individuals allocate the same reproductive effort to bolting, flower and fruit production for fruits that will fully mature or not, yet the fitness benefit is not the same (no viable seeds are produced when fruits fail). We agree that failed fruits contribute to fitness in the sense that individuals that produce fruits that fail have lower fitness than those that produce successful seeds. Whether fruits mature or not does not change the reproductive effort that went into producing these structures. Yet, if resources allocated to fruit production cannot be reabsorbed during fruit abortion, the reproductive effort invested in failed fruits would be wasted. In this case, producing fewer flowers in the first place - or perhaps not even bolting - would optimize resource use and allow allocation of resources to growth and survival. Additionally, fruit abortion occurs late in the season, when plants have already invested considerable reproductive effort in bolting, flower, and fruit development; that is, fruit abortion represents substantial wasted reproductive effort. If fruit abortion depresses individual longevity, and the probability of reproducing or fecundity in the future, the optimal strategy would avoid excess flower and fruit production, because resources may not be reabsorbed and reallocated. We are testing this theory in our system by examining (a) the combined effects of successful and failed reproduction on future fitness components (Figs. 1, 2, 4), (b) the effect of the proportion of the wasted reproductive effort (proportion of fruits that failed) on lifetime fecundity (Fig. 3a-e), and (c) the interactive (synergistic) effect of successful and failed reproduction on lifetime fitness (Fig. 3f-j).

Since we monitored fruit production intensively in the field, we were able to distinguish between successful fruits that matured and failed fruits that did not set seed. From our reading of the literature, very few field studies have quantified both mature and failed fruits, and we think it is important to consider how fruit failure contributes to life history trade-offs, such as the cost of reproduction. We clarified these ideas extensively in the introduction (L. 90-104).

In several places, the effects of treatments are referred to as effects of climate change (e.g. on lines 102 and 149). I do not think that this is correct, and that the authors need to clearly distinguish between effects of treatments and possible implications of these in a climate change context. Moreover, the references to effects of climate change do also sometimes appear to be a bit poorly motivated, e.g. on lines 63-64.

Response: We agree more caution was needed here. We edited the entire manuscript to avoid overstating the validity of our experimental treatments in simulating climate change, and interpret our results more cautiously. As we explained in the methods section (L. 161-163), in this region, early snow removal simulates several factors predicted to change under global change (i.e., decreased winter snowpack, early spring snowmelt, spring frost exposure, decreased water availability during the growing season), and is qualitatively more suitable than most single factor manipulations (temperature or drought alone) to accurately simulate climate change effects.

We also strengthened the claim that few studies have directly investigated effects of natural or experimentally manipulated climate change factors on costs of reproduction, and

added more references. The entire 3rd paragraph of the introduction was revised to better reflect the literature (L. 75-89).

The results section is a bit hard to follow as it is not clear how the results relate to the main predictions, and would suggest the authors to focus more explicitly on the tests of the predictions in the result section. It is also a bit difficult to grasp all the significant two- and three-way interactions, some of which seem hard to find a good biological explanation for (e.g. lines 228-237). The structure of the results and discussion sections should also be better aligned.

Response: Thank you for helping us improve the manuscript. We extensively revised the introduction and the methods to clarify our predictions (L. 105-126), define the fitness components (L. 113-115, 177-180), and strengthen our description of the analytical approach as well as the statistical models (L. 113-118, 183-189). For example, in the methods, we additionally specified how we examine costs of reproduction and how the two- and three-way interactions inherent to our models are to be interpreted (L. 177-180, 186-189). In addition, we revised the results section to indicate how each result links back to the predictions (L. 264-266, 277-279, 284-286, 294-296, 300-301, 306-308, 317-319).

Moreover, we considered aligning the structure of our aims, results and discussion section, but believe this would not necessarily clarify our manuscript. We believe the results section is more comprehensive if broken up by fitness components to reflect our analyses (as indicated by our subheadings). However, in the discussion, we then consolidate the patterns observed across fitness components to provide a comprehensive view of how our climate change simulations shift costs of reproduction and how these shifts may affect life history evolution.

Specific comments:

Lines 29-30, and in the main text: Perhaps this is a matter of taste, but I would not say that life history theory posits a cost of reproduction, but rather that the presence of costs is an empirically based pattern. Life history theory does, however, posit how this cost will influence life history evolution in different environments.

Response: Good point! We edited this for accuracy in the abstract (L. 29-31), and at the beginning of our introduction (L. 47-48).

Line 31, and in the main text: Clarify how trade-offs are altered - through changes in costs or benefits?

Response: We replaced "trade-offs" by "future-fitness costs" (L. 31).

Line 37, and in the main text: What do you mean by "novel" cost?

Response: We previously used the term "novel costs" to define costs that emerged under novel environmental conditions stemming from our snow removal manipulation (i.e., present in snow removal but absent/undetected under benign conditions). However, we understand the term is unclear and we now avoid it altogether. We still refer to "novel environmental conditions under climate change", and we clarified that we examine whether our climate change simulations "induce variation in resource allocation that diverge from optimal allocation strategies, thereby heightening or generating new costs of reproduction under stressful conditions" (L. 78-81).

Lines 76-77: I don't think that this statement is correct, and that it leads to an overstatement of the novelty of the current study in this respect.

Response: We have rephrased this here (L. 90-104) to better reflect that while studies generally quantify the reproductive effort, they rarely distinguish between the portion of the reproductive effort that contributes to future fitness (successful reproduction) and the portion of the effort that may be wasted and could have been saved under more optimal resource allocation (failed reproduction). As explained above in more detail, we also clarified that few field studies (to our knowledge) have tested whether fruit abortion is an adaptive strategy or a waste of resources that could exacerbate costs of reproduction (L. 100-104).

Line 83: Unclear to me why fruit abortion should exaggerate costs.

Response: We considerably revised this section of the introduction (L. 90-104) to better explain how fruit abortion could exaggerate costs of reproduction if these resources cannot be resorbed and reallocated, and if these resources could have been saved in the first place by avoiding excess flower and fruit production.

Line 83-85: The meaning of this sentence is unclear to me. Whether fruit abortion is associated with a strategy or not would hardly influence the costs!?

Response: Again, we hope the revised section L. 90-104 clarifies our rationale.

Lines 92-93: I don't consider this to be a valid argument for why experimental manipulations were not appropriate in this study. As long as manipulations alter reproductive effort, they could still be useful.

Response: We revised the paragraph that introduces our study and experimental approach and removed this particular argument. (L. 109-118, 119-126).

Lines 101-103: I think it would be better if you explained what effects of treatments you expected and how you expected these treatments to interact with site.

Response: We clarified that we expect our early snow removal, which simulates climate change conditions, to exaggerate costs of reproduction across the elevational gradient (L. 108-109, 119-126).

Lines 110-111: I would argue the other way; if fecundity depends on pollinator availability and is more variable, then this would result in larger variation in resource allocation, and in increasing possibilities to detect costs.

Response: We clarified our argument (L. 133-135). The independence from pollinator availability makes it easier to detect shifts in costs of reproduction in response to changes in abiotic factors (without confounding effects of biotic interactions).

Line 150: I am not familiar with the concept of "failed reproductive effort". As reproductive effort is often used to describe the fraction of available resources that are used for reproduction, the concept does not really make sense to me. Reproduction can fail but not reproductive effort.

Response: We agree that this term was not appropriate and replaced it with "failed reproduction", "failed fruit/siliques", and "fruit loss/abortion" as appropriate throughout the manuscript.

Referee: 2

Comments to the Author(s)

The article “Costs of reproduction under experimental climate change across elevations in the perennial forb *Boecheera stricta*” by Hamann, Wadgyman, and Anderson is an ambitious common garden experiment conducted over six years at five locations along an altitudinal gradient in the US Rocky Mountains. It included snow removal treatments at all gardens, which is an impressive undertaking, given these altitudes and the length of time involved. The results are an important contribution to our understanding of how climate change and drought impact reproduction in a perennial montane/alpine plant species. However, I felt there were some areas of the manuscript that needed further clarification or detail. The Introduction lacked specifics about introducing the measures used in this study, while the Discussion introduced concepts related to the evolution of pereniality, give trade-offs to reproduction. More details of the specific types of costs should be highlighted in the Intro, as well as their potential role in shaping evolution.

Response: We thank Referee # 2 for carefully reading our manuscript, for highlighting the large scale of this experiment, and for the constructive criticism. We appreciate that certain aspects of the paper lacked clarity or detailed information, and to address these comments we thoroughly revised the manuscript. We address each comment in more detail below.

Introduction

Minor comments:

Line 55: Introduce the word iteroparous, because it is used later without having been contrasted with the use of semelparous in Line 54.

Response: Thank you for noticing this. We updated the manuscript to introduce the word iteroparous at L. 52.

Line 85: What is meant by bet-hedging? I can guess, but further explanation will help those readers less familiar with allocation/trade-off literature.

Response: We edited this sentence and removed the term "bet-hedging" altogether (L. 61-63).

Major comments:

The costs of reproduction referenced in other studies should introduce your own approach for quantifying costs (2nd and 3rd paragraphs). I found the Introduction lacked specifics about these costs, and was at times repetitive. For example, the sentences that start on lines 57 and 59 essentially make similar points. Furthermore, the 2nd and 3rd paragraph of the Intro were more or less the same. In both cases, there were references indicating that costs of reproduction increased under some scenarios, but these costs were not defined and explained. Any indication how these costs impacted future growth or reproduction? How do these relate to your study? You look at future fecundity and survival, so please establish the motivation for your measurements. In the description of tests, on Line 98, it is predicted that “novel” costs of reproduction emerge. What do you mean by “novel?” Please contrast with previous studies (this ties in with the

comments above). Also, this will help prepare readers the methods, where you introduce “longevity costs,” “fecundity costs,” etc.

Response: Thank you for these thoughtful comments that helped us strengthen our introduction. We revised the entire introduction to avoid repetition while clarifying the theoretical background, better defining our terminology and measured variables, and hopefully, express our aims more clearly. Specifically, the second paragraph now introduces resource acquisition vs. allocation and describes how the expression of costs of reproduction depends on resource availability and abiotic conditions (L. 54-74). Additionally, we outline our experimental and analytical approaches that accounts for microenvironmental variation in resource availability and acquisition (L. 105-118), which also allows us to highlight the strength of our complementary approach consisting of examining both phenotypic and genetic correlations between initial and future reproduction (L. 115-118).

Additionally, we were careful to define our terms more clearly. For example, we now avoid the term "novel costs", but examine whether simulated climate change can " induce variation in resource allocation that diverge from optimal allocation strategies, thereby heightening or generating new costs of reproduction under stressful conditions " (L. 78-81), and whether "reproduction under novel - often stressful - conditions could come at greater fitness costs" (L. 86-89). We also clarified how the initial reproductive effort can affect or be detected in several future fitness components such as survival, growth, future probability of reproduction, future fecundity (L. 90-92, 113-115, 177-180, 186-189).

Methods

Minor comments:

Line 120: How many blocks per garden?

Response: We added this information in parenthesis (L. 147-148). The number of blocks range was 14 in the two lowest elevation gardens (2553m and 2710m), 16 in the mid-elevation gardens (2890m and 3133m) and 7 in the highest garden (3340m). Logistical constraints prevented us from including more blocks in the highest garden. Blocks were evenly divided between snow removal and control treatments, except in the highest garden where 4 blocks were allocated to control and three to snow removal.

Line 126: It took me a bit to understand that these supplementary tables are from “reference” 40. Please clarify. Also, how do the six years of your study compare with historical averages of temperature and snowfall? An important aspect of long-term studies like this is that it can encompass a range of variation across years.

Response: Thank you for pointing this out, but we feel this was already clear as the citation explicitly said "Table S2 and S3 IN 40" (L. 150-153). The alternative would be to refer to the paper without the SI Table detail, but we hope this actually helps point readers to the relevant data. Additionally, we clarified that our "multiyear experiment captured inter-annual climate variation representative of historical climates" and now refer to "[Table S1-3 and Fig. S1 IN REFERENCE 41]" L. 152-153.

Those supplemental files of Anderson and Wadgymar (reference 41 in the manuscript) provide data on climatic variation in reference to historical levels. In this response letter, we are pasting in Fig. S1 of Anderson and Wadgymar (2020). For that study, we implemented two additional treatments to evaluate seed germination that we do not have in the current manuscript (snow addition, and snow addition plus supplemental watering). For Fig. S1a below, please note

that snowmelt timing in our control treatment (in blue) and snow removal treatment (in red) at the mid-elevation garden (2890m) fall within the range of historical variation in snowmelt timing at the Rocky Mountain Biological Laboratory (in black):

Line 127: Plant diameter. Do you mean rosette diameter? Stem diameter?

Response: We edited this to "rosette diameter" (L. 154).

Line 155: Instead of using "garden" as a variable, how about referring to it as "altitude?"

Response: We treated "garden" as a categorical fixed factor in our statistical models. We are concerned that replacing "garden" with "elevation" could lead some readers to the erroneous conclusion that we treated garden elevation as a continuous predictor in our models. In the revision, we replaced the term "garden" with "garden elevation" throughout the manuscript. This replacement allows us to clearly distinguish the elevation of the garden from the elevation of the source populations (for example L. 184, 224, 231, 244, etc.). However, we prefer the term "garden" because it reflects the true nature of the experiment and the statistical analyses.

Line 163: "Geographic distance". Is this again the difference between "source" and "elevation?"

Response: Yes, we incorporated two metrics to account for local adaptation: the elevational distance between source population and transplant garden (captures the difference in elevation and can be positive or negative depending on whether source populations are transplanted to higher or lower elevation sites), and the geographical distance between source population and transplant garden (captures the distance as the crow flies between sites, measured in meters). We edited for clarity (L. 194-196).

Line 165-166: Longevity costs. Please define what you mean by longevity costs. Also, in the

caption for Figure 1, you describe what ratios that are $>$ or $<$ 1 indicate. Please describe here, as well.

Response: In the revised manuscript, we clarified in the introduction that we assess costs of reproduction in several fitness components, including longevity, future reproductive odds, and future fecundity (L. 113-115, 177-180). At L. 179 and 200-203, we indicate that longevity is plant lifespan, calculated as the number of elapsed days from planting to death or the last census data. For each of these fitness components, costs are detected when high allocation to the first-year reproductive effort decreases subsequent fitness, and variation across elevation and treatments is examined via the 2- and 3-way interactions in our statistical models (L. 186-189).

Additionally, we specified that "Hazards ratios >1 or <1 indicate that high initial reproduction increased or decreased mortality, respectively (i.e., reduced or increased life span)." (L. 200-201).

Major comments:

Fitness and fecundity aren't always clearly distinguished. I think the intent was to use fitness for 2014, and then fecundity for the rest of the experiment, but this was not always clear. If I'm correct about that interpretation, I think specifying "first-year fitness" and "lifetime fecundity" will help.

Additionally, the section on "lifetime costs" starting on line 188 appeared similar to the section on "fecundity costs" on the previous page. An introductory sentence for each paragraph should indicate what each test reveals. (The first sentence on lines 172-173 doesn't tell me).

Response: Thank you for pointing out these terms and sections needed attention. We carefully revised the manuscript to clarify these terms and use them more consistently as per your suggestion. Specifically, in L. 113-115, 177-180, we indicate that we examine costs of reproduction across several fitness components to avoid repetitions later.

Indeed, we use the term "first-year fitness" when referring to our measure of first-year successful reproduction. At line 180-182, we state: "We calculated first-year fitness as the total mature silique length produced by each individual in 2014, with values of zero for individuals that produced no mature fruits." The term fitness incorporates whether a plant reproduced or not, and - if it reproduced - how fecund it was.

As explained in the "fecundity costs" section (L. 206-220), fecundity costs are analyzed with hurdle models, which first examines how the first-year reproductive effort affects the future probability of reproducing, and then for reproductive individuals only, their fecundity. We define that future fecundity at L. 213-215 as "the sum of mature silique length produced from 2015-2019, excluding individuals that did not reproduce." It focuses on the silique length produced after the first year and excludes individuals that did not reproduce. We adopted this hurdle model approach because fitness from 2015-2019 was zero-inflated (i.e., zero values were overrepresented, as is often the case in field fitness data when not all individuals successfully reproduce). Our probability of reproduction logistic regression models assesses future fitness costs of all individuals (including those that remained in a vegetative state), and our fecundity gamma regression models then evaluated differences in fecundity amongst individuals that produced mature fruits.

Finally, "lifetime fecundity" is defined L. 226-229 as "the sum of mature silique length from 2014-2019 for individuals that flowered in at least one growing." Again, we exclude non-reproductive individuals, but this time we sum the silique length produced over the entire

experimental period and thus the lifetime of the individuals, which reveals the contribution of the allocation to the first-year reproductive effort on lifetime fecundity.

Results:

The paragraph starting on Line 233 is not necessary in its entirety, since it more or less reports the same results as the previous paragraph.

Response: This section reports on the interaction between garden elevation and first-year failed reproduction, while the previous section describes the interaction between garden elevation and first-year successful reproduction. The patterns across variables are different and should remain distinct (L. 270-274).

Line 242-243: When you state “significant cost of reproduction,” indicate that you mean a cost in terms of future reproduction, since you are examining multiple different “costs”.

Response: We defined "costs of reproduction" in the introduction and then more specifically in methods (L. 48-49, 78-81, 113-115, 177-180, 186-189) to avoid repetitions throughout the manuscript. While we sometimes talk about general costs of reproduction, we most often refer to specific longevity costs, fecundity costs, or lifetime costs. Furthermore, we break up our result section into the different fitness components that we examined so that it remains transparent for which fitness component these costs were documented.

Line 247: By “environmental context,” do you mean elevation?

Response: In this instance (L. 284), we clarified that we mean both garden elevation and our experimental treatment. Please also see our response to your comment below (relating to L. 288).

Line 267: “The synergistic effect of successful and failed reproductive effect...” This sentence was unclear.

Response: We have clarified this issue here and elsewhere. By synergistic, we referred to statistically significant interactions of successful and failed reproduction. We now refer to "interactive" or "synergistic" effects of successful and failed reproductive effects. Specifically, our models include both successful fitness (silique length), failed reproduction (number of failed siliques), elevation and treatment as additive terms and multiplicative terms via their interactions. In this instance (L. 306-314), we found a significant three-way interaction between successful first-year fitness, first-year failed reproduction, and garden elevation – so both successful and failed reproduction in the first year exert an interactive effect (or act synergistically) on lifetime fitness (and this effect differs across elevations).

Discussion:

Line 288: Again “environmental context” should be clarified. Elevation? Precipitation? Also, I don’t think you’ve previously discussed the evolution of perenniality, so that statement isn’t the best way to introduce the discussion. This comes up again later in the Discussion (last page), and should be addressed more in the Introduction.

Response: We agree the term may sound vague, but at this junction in the discussion, we aim to emphasize that costs of reproduction depend on environmental conditions instead of naming specific abiotic factors that may affect these costs. To keep a broader opening here, we

edited this text to "environmental conditions" (L. 331). Elsewhere, when appropriate, we refer to specific environmental factors such as elevation or snow removal/drought.

Additionally, we now introduce more thoroughly the concept of evolution of life history strategies in association with shifts in costs of reproduction under changing environmental conditions (L. 78-81, 107-109, 121-124).

Line 311: "Reduced evapotranspiration." Do you have data for this? Or was this from a prior publication?

Response: Yes, we do have data and we refer to the supplementary data from a previous publication to point readers towards this data (Table S1 in reference 41) L. 355.

Line 312: Do you mean "shorter" rather "shortened"?

Response: Yes, thank you. We edited to "shorter" L. 354.

Line 315: Expand on the hypothesis regarding variation in resource acquisition. How is this helping plants more at some elevations vs. others?

Response: As per your and the first reviewer's suggestions, we have clarified the distinction between variation in resource allocation versus variation in resource allocation (L. 54-74). In the revision, we indicate that under control conditions, the mid-elevations have the mildest climate, with the highest water availability without being limited by short growing season as in higher elevations (L. 353-355). Thus, individuals have access to more resources to allocate to reproduction, and may even be able to continue to grow during reproduction to compensate for excessive flower and fruit production. We strengthened our rationale L. 355-362 and 396-407.

I like introducing the concept of how the variation of these trade-offs over elevations may shape evolution, but some context for this should be included in the Introduction.

Response: We appreciate that we needed to introduce this concept earlier. In this revised introduction, we state that costs of reproduction underlie the evolution of life history strategy (L. 50-51). We develop this concept broadly (L. 78-81) and then more specifically in the context of our alpine study system (L. 107-109), by saying that "We predict that climate change could shift costs of reproduction and drive the evolution of alternative life history strategies in this system."

References

1. van Noordwijk A.J., de Jong G. 1986 Acquisition and allocation of resources: their influence on variation in life history tactics. *The American Naturalist* **128**(1), 137-142. (doi:10.1086/284547).
2. Sletvold N., Ågren J. 2015 Climate-dependent costs of reproduction: Survival and fecundity costs decline with length of the growing season and summer temperature. *Ecology Letters* **18**(4), 357-364. (doi:10.1111/ele.12417).
3. Euler T., Ågren J., Ehrlén J. 2012 Floral display and habitat quality affect cost of reproduction in *Primula farinosa*. *Oikos* **121**(9), 1400-1407. (doi:10.1111/j.1600-0706.2012.20433.x).